# Ligustilide Inhibits Tumor Angiogenesis by Downregulating VEGFA Secretion from Cancer-Associated Fibroblasts in Prostate Cancer via TLR4

**DOI:** 10.3390/cancers14102406

**Published:** 2022-05-13

**Authors:** Jing Ma, Xu Chen, Yumo Chen, Ning Tao, Zhihai Qin

**Affiliations:** 1Key Laboratory of Protein and Peptide Pharmaceuticals, Institute of Biophysics, Chinese Academy of Sciences, Beijing 100101, China; majing@ibp.ac.cn (J.M.); chenxu191@mails.ucas.ac.cn (X.C.); 2The First Affiliated Hospital of Zhengzhou University, Zhengzhou University, Zhengzhou 450052, China; 3College of Life Sciences, University of Chinese Academy of Sciences, Beijing 100101, China; 4Beijing National Day School, Beijing 100141, China; chenyumo2005@126.com

**Keywords:** ligustilide, cancer-associated fibroblasts, angiogenesis, vascular endothelial growth factor, Toll-like receptor 4

## Abstract

**Simple Summary:**

Inhibiting the production of vascular endothelial growth factor A (VEGFA) can inhibit angiogenesis, thereby inhibiting tumor growth. We found that ligustilide can inhibit the secretion of VEGFA from prostate cancer-related fibroblasts (CAFs), and this signaling pathway is related to Toll-like receptor 4 (TLR4)-ERK/JNK/p38. Using the above receptors or signals pathway molecule blockers can block the effect of ligustilide to downregulate the secretion of VEGFA from CAFs. The concentration of ligustilide used in this study does not directly inhibit the growth of CAFs, but changes its function, and this study is different from the direct blocking effect of the existing VEGFA antibodies, but cuts off the source of VEGFA, which is expected to become a novel therapeutic strategy for oncology.

**Abstract:**

CAFs secrete VEGFA in the tumor microenvironment to induce angiogenesis and promote tumor growth. The downregulation of VEGFA secretion from CAFs helps block angiogenesis and exerts an anti-tumor effect. In vivo experiments showed that the angiogenesis of the tumor-bearing mice in the ligustilide group was significantly reduced. The results of MTT, tube formation, Transwell and scratch experiments showed that ligustilide did not affect the proliferation of HUVECs in a certain concentration range (<60 μM), but it inhibited the proliferation, tube formation and migration of HUVECs induced by CAFs. At this concentration, ligustilide did not inhibit CAF proliferation. The qPCR and WB results revealed that ligustilide downregulated the level of VEGFA in CAFs via the TLR4-ERK/JNK/p38 signaling pathway, and the effect was attenuated by blockers of the above molecules. Ligustilide also downregulated the autocrine VEGFA of HUVECs induced by CAFs, which inhibited angiogenesis more effectively. In addition, ligustilide inhibited glycolysis and HIF-1 expression in CAFs. Overall, ligustilide downregulated the VEGFA level in CAFs via the TLR4-ERK/JNK/p38 signaling pathway and inhibited the promotion of angiogenesis. This study provides a new strategy for the anti-tumor effect of natural active molecules, namely, blockade of angiogenesis, and provides a new candidate molecule for blocking angiogenesis in the tumor microenvironment.

## 1. Introduction

Angiogenesis is one hallmark of tumors [1]. Tumor growth requires large amounts of nutrients and oxygen, and angiogenesis becomes indispensable [2]. The presence of tumors induces angiogenesis and tumor growth is controlled by surrounding blood vessels. Therefore, anti-angiogenesis induces tumor dormancy, and angiogenesis promotes tumor growth. Therefore, anti-angiogenesis has become a therapeutic strategy for tumors [3].

Vascular endothelial growth factor (VEGF) exhibits direct pro-angiogenic functions and stimulates angiogenesis by recruiting stromal cells that support angiogenesis and secrete VEGF [4]. VEGF promotes angiogenesis and increased vascular permeability and extracellular matrix degeneration [5,6].

VEGF antibodies achieved good efficacy as an anti-tumor therapy [7,8], but it may cause side effects, such as gastrointestinal perforation and nephrotic syndrome [9,10]. Not all patients are sensitive to these antibodies, and some patients have drug resistance [11]. Drug resistance may be related to the tumor microenvironment [12]. Therefore, reducing the production of VEGFA from the source may be better than blocking VEGFA for anti-tumor therapy.

Cancer-associated fibroblasts (CAFs) are stromal cells in the tumor microenvironment that secrete cytokines, such as VEGF, hepatocyte growth factor (HGF) and fibroblast growth factor (FGF), to promote angiogenesis [13], and IFN-γ downregulates VEGF secretion from CAFs, blocks angiogenesis, and inhibits tumor growth [14]. However, IFN-γ may lead to immune escape [15]. Therefore, screening for an active molecule that downregulates VEGF secretion from CAFs without causing immune escape should be an anti-tumor drug candidate.

Ligustilide is the main component of the volatile oil of the traditional Chinese medicine Angelica sinensis [16], and it has inhibitory effects on non-small-cell lung cancer [17], osteosarcoma cells [18], ovarian cancer cells [19] and malignant glioma cells [20]. Ligustilide induced apoptosis in prostate cancer-associated fibroblasts [21], and the molecule restored the proliferation of T cells inhibited by CAFs [22]. Therefore, whether this molecule downregulates VEGF in CAFs to inhibit angiogenesis is not clear. The present study used ligustilide on CAFs, and the secretion of VEGF and pro-angiogenesis were detected to elucidate the anticancer mechanism of ligustilide.

## 2. Results

### 2.1. Ligustilide Reduces Blood Vessel Density in Prostate Cancer Tissue

Ligustilide is an extract of the Chinese herbs Angelica and Chuanxiong. A previous study found that ligustilide significantly inhibited tumor growth in a prostate cancer (RM-1)-bearing mouse model. The present research established a subcutaneous prostate cancer-bearing model (Figure 1A), and ligustilide significantly reduced the expression levels of α-SMA and CD31 in tumor tissue, which represented CAFs and vascular endothelial cells, respectively (Figure 1B). The results showed that the α-SMA, VEGFA and HGF transcriptional levels in the tumor tissue of the ligustilide-treated group were significantly reduced compared to the control group (PBS) (Figure 1C). In vivo experiments showed that ligustilide significantly reduced vascular density in prostate cancer tissue.

### 2.2. Ligustilide Significantly Inhibits the Pro-Angiogenesis Effect of CAF Supernatant

To explore how ligustilide inhibited angiogenesis in tumor tissue, an in vitro cell model was constructed. Prostate cancer-associated fibroblasts (CAFs) were pretreated with ligustilide, and the supernatant of CAFs without ligustilide was collected (Figure 2A). Ligustilide directly had no significant effect on the proliferation of human vascular endothelial cells (HUVECs) (Figure 2B). CAF supernatant promoted the proliferation, migration and tube-like structures of vascular endothelial cells, and ligustilide inhibited these effects. However, the NAF supernatant had no similar effect on HUVECs (Figure 2C–I). Ligustilide significantly inhibited the pro-angiogenesis effect of CAFs, and NAFs had no significant effect on angiogenesis. The concentration of ligustilide (0–40 μM) that inhibited the pro-angiogenic effect of CAFs did not inhibit the proliferation of CAFs (Appendix A). Therefore, ligustilide did not inhibit the proliferation of CAFs but did inhibit the pro-angiogenic effect on vascular endothelial cells. Ligustilide may induce a shift of CAF function to NAFs.

### 2.3. Ligustilide Inhibits the Pro-Angiogenic Effect of CAFs via the TLR4-AP-1 Signaling Pathway

To explore the molecular mechanism of ligustilide inhibition of the pro-angiogenic effects of CAFs, CAFs were pre-treated with blockers of TLR2 or TLR4. The results showed that TLR4 was involved in the inhibitory effect of ligustilide on CAFs, which promoted the proliferation, migration and tube-like structures of HUVECs (Figure 3A–E). However, TLR2 had no apparent effect (Appendix A). Ligustilide promoted the phosphorylation of p38, ERK and JNK via MYD88 to activate the AP-1 signaling pathway, which is downstream of TLR4 (Figure 3F–H). After treatment of CAFs with inhibitors of p38, ERK or JNK, the inhibitory effect of ligustilide on CAFs was significantly attenuated (Figure 3I–M), and ligustilide-induced phosphorylation of p38, ERK and JNK was attenuated (Appendix A). Ligustilide promoted the phosphorylation of p38, ERK and JNK via TLR4 in CAFs, which promoted the expression of the transcription factor AP-1, and ligustilide inhibited the pro-angiogenic effects of CAFs via the TLR4-p38/ERK/JNK-AP-1 signaling pathway.

### 2.4. Ligustilide DownRegulates the Expression Level of VEGFA in CAFs via the TLR4-AP-1 Signaling Pathway

Ligustilide inhibited the pro-angiogenic effect of CAF supernatant. We explored the effects of ligustilide on CAF phenotype, and ligustilide significantly downregulated the expression of VEGFA, α-SMA and S100A4. These effects were blocked when CAFs were pretreated with a TLR4 inhibitor (Figure 4A–D). The effect of ligustilide-induced down-regulation of VEGFA levels was attenuated when CAFs were pretreated with p38, ERK, or JNK inhibitors (Figure 4E). Histochemical results also suggested that ligustilide down-regulated the expression levels of VEGFA (Figure 4F) and MMP9 (Appendix A) in CAFs via TLR4. CAF supernatants upregulated VEGFR2 and downregulated VEGFR1 in HUVECs, and ligustilide-treated CAFs recovered VEFGR1 and downregulated VEGFR2 (Figure 4G). After pretreatment of HUVECs with a VEGFR blocker, the migratory capacity of vascular endothelial cells cultured with the supernatant of CAFs and ligustilide-treated CAFs was attenuated (Figure 4H,I).

### 2.5. Ligustilide Inhibits Glycolysis and HIF-1 Expression in CAFs

Ligustilide significantly downregulated the expression levels of hexokinase (HK1/HK2), glucose transporter (GLUT1), pyruvate dehydrogenase kinase isoenzyme 1 (PDK1) and lactate dehydrogenase (LDHA) in CAFs (Figure 5A) and significantly reduced intracellular lactate (Figure 5B). Ligustilide upregulated Jab1 and P53 of CAFs and downregulated HIF-1 (Figure 5C–E). Ligustilide inhibited glycolysis and HIF-1 expression in CAFs, but it did not affect glycolysis or HIF-1 expression in NAFs (Appendix A).

### 2.6. Ligustilide Attenuates the Signaling Pathway Involved in the Proliferation of HUVECs Induced by CAF Supernatant

To explore the mechanism of ligustilide-pretreated CAF supernatant on HUVECs, the signaling pathways related to HUVECs and angiogenesis were detected. CAF supernatant significantly upregulated the expression of AKT, p38, ERK, AP-1, MMP9 and VEGFA in HUVECs, and ligustilide significantly attenuated the upregulation of CAF supernatant (Figure 6A). CAF supernatant and ligustilide-pretreated CAF supernatant had no significant difference in the transcriptional levels of DLL4, Foxc2 and Notch1 in HUVECs (Figure 6B). The changes in the protein levels of *p*-AKT, *p*-p38 and AP-1 detected in HUVECs were consistent with the mRNA levels (Figure 6C). CAF supernatant significantly activated vascular endothelial cells and angiogenesis-related signaling pathways and effector proteins, and ligustilide pretreatment of CAF supernatant attenuated the activation effect of CAF supernatant on HUVECs (Figure 7).

## 3. Discussion

Compared to MDSCs [23], TAMs [24], Tregs [25], Bregs [26] and TANs [27], which are myeloid-derived immune negative regulatory cells, CAFs are the residual cells in the tumor microenvironment that create a “niche” for tumor cells and participate in the formation of the tumor microenvironment [28]. CAFs secrete collagen to promote the stemness of cancer cells and secrete TGF-β [29] to recruit T cells and promote tumor immune escape. Notably, CAFs promote tumor angiogenesis by secreting VEGFA, and a large amount of VEGFA may result in leaky blood vessels [30] and interfere with the normalization of blood vessels [31] and the inability of immune effector cells to enter the cancer nest. Selective CAF knock out induced ischemic necrosis of some tumor cells. Ischemic and necrotic tumor cells release tumor antigens. After antigen presentation, T cells are activated to destroy tumors [32]. Therefore, ischemic necrosis plays a key role in this process, and VEGFA acts as a switch for ischemic necrosis.

It is better for anti-angiogenesis of VEGFA to reduce the production from CAFs rather than direct inhibition of VEGFA. The current clinical blocking strategy is an after-the-fact measure, i.e., VEGFA is produced and secreted, and angiogenesis is inhibited via antibody blockade. The disadvantage of this method is that the source of VEGFA is not removed, and VEGFA is continuously produced. The continuous use of antibodies leads to high costs, and the blockade of VEGFA in non-cancer nests causes side effects, such as gastrointestinal perforation and proteinuria. Ligustilide precisely reduced the production of VEGFA by CAFs in the cancer nest and may have no effect on VEGFA in normal tissue.

VEGFR2 is the main receptor of angiogenesis [4]. VEGFR1 has a competitive antagonistic effect on VEGFR2 by “decoy” binding to VEGFA [33]. Ligustilide upregulated the VEGFR1 transcription levels that were downregulated by CAFs, but downregulated the VEGFR2 transcription levels that were upregulated by CAFs. This finding indicates that ligustilide inhibits VEGFA-promoted angiogenesis by modulating the balance between VEGFR1 and VEGFR2 induced by CAFs.

TLR4 is an important pattern recognition receptor involved in the bacterial lipopolysaccharide-induced inflammatory response [34]. Notably, TLR4 is also involved in a variety of natural active molecules, such as polysaccharides of Lentinus edodes [35,36] and cinnamaldehyde [37] on CAFs. Asparagus polysaccharides [38], eugenol [39] and curcumin [40] induce apoptosis or modify the function of negative immune cells in the tumor microenvironment, such as MDSCs. Moreover, TLR4 is also involved in the induction of CAF apoptosis [21] and the inhibition of the immunosuppressive function of prostate CAFs by ligustilide [22]. Our results showed that the effect of ligustilide downregulation of VEGFA disappeared with the use of the TLR4 blocker CLI-095, which indicated that TLR4 also participated in the inhibitory effect of ligustilide on the proangiogenic effect of CAFs.

MyD88 is the main signaling molecule in the TLR4 signaling pathway [41]. TLR4-MyD88-TRAF6-TAK1-JNK/p38-AP-1 is involved in the signaling pathway of LPS-induced cytokine production, which transmits extracellular signals into the nucleus [42]. The present study found that ligustilide was involved in this signaling pathway in the process of regulating the secretion of VEGFA from CAFs. ERK is involved in VEGFA-induced angiogenesis [43], and the present study found that ligustilide also inhibited VEGFA production by downregulating ERK in CAFs.

Ligustilide upregulated Jab1 and downregulated HIF-1α simultaneously. The following mechanism is proposed. Jab1 directly binds to HIF-1α, which increases HIF-1α stability [44], and Jab1 also binds to AP-1 [45]. We speculate that ligustilide increases the binding of AP-1 to Jab1 and competitively antagonizes the binding of Jab1 to HIF-1a, which downregulates HIF-1α. Another possibility is that Jab1 activates AP-1 [46] via phosphorylation, and JunD, a member of the AP-1 family, antagonizes HIF-1α [47], which suggests that the activation of AP-1 may also antagonize HIF-1α. The p53 protein also binds to HIF-1α to exert a negative regulatory effect [44]. HIF-1α is a key molecule that promotes VEGF secretion [48], and the downregulation of HIF-1α leads to decreased VEGF secretion and the inhibition of angiogenesis.

## 4. Materials and Methods

### 4.1. Cell Culture

CAFs (PF179T-CAF, isolated from a prostatectomy specimen, marginal to prostate cancer, hTERT immortalized; designated PF179; 179, patient number, Department of Urology, University of Innsbruck, Innsbruck, Austria) and NAFs (isolated from a prostatectomy specimen, normal tissue of prostate) were provided by Professor Ju Zhang at the College of Life Sciences, Nankai University. HUVECs (human umbilical vein endothelial cells) were a gift from Professor Yan Xiyun at the Institute of Biophysics, Chinese Academy of Sciences. RM-1 (murine prostate cancer cell line) was purchased from the Cell Resource Center of Shanghai Institutes for Biological Sciences. All cells were cultured in DMEM (HyClone, South Logan, UT, USA) supplemented with 10% FBS (PAN, Adenbach, Bavaria, Germany) and 1% penicillin/streptomycin at 37 °C in an incubator with 5% CO_2_.

### 4.2. Preparation of Ligustilide Solution and CAF Supernatant

A 1 M stock solution in DMEM containing 1% ethanol of ligustilide (Sichuan Weikeqi Biotechnology Co., Ltd., Chengdu, China) was prepared and stored at −20 °C.

CAFs were seeded in 24-well plates and cultured with 1 mL of DMEM, 2 × 10^4^ cells/well. After the cells adhered overnight, they were treated with different concentrations of ligustilide (0, 10, 20 and 40 μM) for 48 h. The medium containing ligustilide was discarded and replaced with fresh medium for 24 h. The supernatant of CAFs was collected and filtered using a 0.45-μm filter.

### 4.3. Cell Viability Assay (MTT)

The cytotoxicity of ligustilide was determined using a quantitative cell viability assay and MTT (3-[4,5-dimethylthiazol-2-yl]-2,5-diphenyl-tetra-zolium bromide, Amresco, Washington, WA, USA) assay. Cells were seeded in 96-well plates at a density of 5 × 10^3^ cells/well and incubated with ligustilide at different concentrations for 24 h at 37 °C in a CO_2_ incubator. Next, 10 μL of MTT (5 mg/mL) was added to each well and incubated for 3–4 h in the dark at 37 °C. Triple combination buffer (10% SDS, 5% isobutanol, 0.012 mol/L HCl, dissolved in distilled water; 100 µL) was added to dissolve the crystals overnight. A microplate reader (BIO-RAD Laboratories, Hercules, CA, USA) measured the absorbance at 570 nm.

### 4.4. Real-Time Quantitative Polymerase Chain Reaction (RT-qPCR)

Total RNA was extracted after lysis of cells or tumor tissue with TRIzol (Invitrogen, Carlsbad, CA, USA) and quantified using an ND-1000 spectrophotometer (NanoDrop Technologies, Wilmington, DE, USA). Equal amounts of total RNA (1 μg) from samples were reverse transcribed into cDNA using PrimeScript RT Master Mix (Takara, Kusatsu, Shiga, Japan). Gene expression quantification was performed using RT-qPCR and a SYBR Green II qPCR kit (GenStar, Beijing, China). Details of the primers used for RT-qPCR assays are provided in Appendix A.

### 4.5. Western Blot Assay

CAFs or HUVECs treated under different conditions were lysed with RIPA solution (Beyotime, Shanghai, China) to obtain total protein. Lysates were placed on ice for 30 min and centrifuged at 12,000 rpm for 30 min. The supernatant was subjected to a BCA protein assay (Thermo Fisher Scientific, Waltham, MA, USA) to determine the protein concentration of each sample. Protein samples (20 μg/sample) were separated in a 10% SDS polyacrylamide gel under denaturing conditions then electrotransferred onto nitrocellulose membranes (GE Healthcare, Milwaukee, WI, USA) for 70 min at 100 V. The membranes were blocked with 3% BSA in PBS-T (0.1% Tween-20) for 90 min at room temperature and incubated with diluted primary antibodies overnight at 4 °C. The following primary antibodies were used: *p*-TAK1(5206S), TAK1(5206S), Jab1(6895S), HIF-1α(36169S), AKT(2920S), MyD88(4283S), AP-1(3270S), *p*-p38(4511S), P38(8690S), *p*-JNK(4668S), JNK(9252S), *p*-ERK(4370S), ERK(4695S), p53(2524S), α-SMA (ab5694), VEGFA(19003-1-AP), and S100A4(ab218512). After three washes in PBS-T for 30 min, the blots were incubated with horseradish peroxidase (HRP)-conjugated secondary antibodies: anti-rabbit IgG (AS014) and anti-mouse IgG (AS003) and developed using a chemiluminescence imaging system (Clinx Science Instruments Co., Ltd., Shanghai, China). An HRP-conjugated anti-GAPDH antibody (AC035) was used as an internal loading control.

### 4.6. ELISA

A VEGFA ELISA kit (RK00023) was used to detect the VEGFA concentration in CAF supernatant treated with ligustilide. Samples or standards (100 μL) were added to pre-coated well plates and incubated overnight at 4 °C. After removing the sample, 300 μL/well washing buffer (wash 5 times, 1 min each time) was added. The washing buffer was removed, and 100 μL/well of working biotin-conjugated antibody was added and incubated for 1 h at 37 °C. After incubation, the cells were washed 5 times, 100 μL/well streptavidin-HRP was added, and the cells were incubated for 30 min at 37 °C. After washing, 100 μL TMB substrate was added to each well and incubated for 15 min at 37 °C in the dark. A stop solution (50 µL) was added for 5 min, and a microplate reader (BIO-RAD Laboratories, Hercules, CA, USA) was used to read the absorbance at 450 nm.

### 4.7. Cell Migration Assays

HUVECs (3 × 10^5^ cells/well) were plated in 6-well plates. When the HUVECs grew to 80% confluence, the monolayer was scraped using a 10-μL pipette tip. The cells were washed 3 times with PBS to remove detached cells and cultured in different conditioned media. The scratches were recorded at different time points, and the width of the gap was calculated and measured using ImageJ software.

HUVECs (2 × 10^4^ cells/well) were seeded in 24-well plate chamber inserts (Corning Life Sciences, cat. no. 3422). Culture medium (300 μL) was added to the chamber, and 700 μL of culture medium (containing 30% supernatant of CAFs or NAFs treated with different conditions) was added to the chamber. After culturing at 37 °C for 6 or 12 h, the chamber was removed, and the cells in the upper layer of the chamber were removed with a cotton swab. Cells were stained with 0.5% crystal violet and washed 3 times with PBS after 30 min of staining. After taking pictures with a Nikon inverted microscope (Ti-S), ImageJ was used to count the number of cells passing through the chamber.

### 4.8. Tube Formation Assay

Tube-like structures were generated by seeding HUVECs (1 × 10^4^ cells/well) on Matrigel (BD Biosciences, Franklin Lakes, NJ, USA) (80 µL/well) in a 96-well plate at 37 °C in a 5% CO_2_ incubator. Images of the tubular structures were taken using a Nikon inverted microscope (Ti-S).

### 4.9. Immunohistochemical Staining (IHC)

Frozen sections of prostate cancer tissue from mice were taken. The frozen sections were air-dried and fixed in −18 °C pre-cooled acetone at room temperature for 15 min then placed in a fume hood for 2 h. After fixation with 4% paraformaldehyde (PFA) for 5 min, the cells were washed 3 times with PBS and blocked with 3% BSA for 2 h. Sections were placed in a humidified chamber, and incubated overnight at 4 °C with α-SMA (ab5694) and CD31 (550274) primary antibodies. Following three washes, the samples were treated with secondary antibody and DAPI (Invitrogen, Carlsbad, CA, USA) for 1 h, fully rinsed with PBS and examined at room temperature. After several washes, the slides were photographed using a microscope (OLYMPUS, Tokyo, Japan).

### 4.10. Immunofluorescence

CAFs were seeded in 12-well plates at a density of 2 × 105 cells/well. After 24 h, 40 μM of ligustilide at was added for 24 h. The cells were washed twice with PBS and fixed with 4% PFA for 10 min. The cells were fully washed and treated with 0.1% Triton X-100 in PBS for 5 min. The cells were blocked with 3% BSA at room temperature for 30 min and stained with 1:200 diluted primary antibody overnight at 4 °C. After several washes, the samples were incubated with 488 donkey anti-rabbit IgG (Invitrogen, USA) for 60 min at 37 °C. After 3 washes, DAPI was used for nuclear counterstaining. The samples were rinsed with PBS and photographed using a digital microscope (OLYMPUS, Tokyo, Japan).

### 4.11. Lactate Detection

Glycolysis was assessed using a glucose metabolism-associated lactate assay kit (Solarbio, Beijing, China) following the manufacturer’s protocol.

### 4.12. Mouse Tumor Models

Wild-type C57BL/6 mice were obtained from the Weitonglihua Company (Beijing, China). All mice were kept in a specific pathogen-free environment at the Institute of Biophysics, Chinese Academy of Sciences. A total of 5 × 10^5^ RM-1 cells were injected subcutaneously into the left lower abdomen of C57BL/6 mice to establish a subcutaneous tumor-bearing model. These mice were divided into the ligustilide group and PBS group, i.e., the control group, with 6 mice in each group. The mice were intraperitoneally injected with ligustilide (5 mg/kg) beginning on the second day after tumor bearing. After 18 days of tumor bearing, the mice were sacrificed, and the tumor tissue was removed for analysis.

### 4.13. Statistical Analysis

All experiments were independently repeated at least three times. To compare two columns of data, Student’s *t*-test was used. To compare multiple column data, we performed a one-way ANOVA. To compare multiple groups of data, we performed a two-way ANOVA. Data were analyzed using GraphPad Prism 7.0 software. Statistical significance was determined with values of *p* > 0.05 (ns), *p* ≤ 0.05 (*), *p* ≤ 0.01 (**) and *p* ≤ 0.001 (***), which represent non-significant, significant, very significant and highly significant, respectively.

## 5. Conclusions

Ligustilide downregulated the VEGFA level in CAFs via the TLR4-ERK/JNK/p38 signaling pathway and inhibited the promotion of angiogenesis. The effect related to the downregulation of HIF-1α.

## Figures and Tables

**Figure 1 cancers-14-02406-f001:**
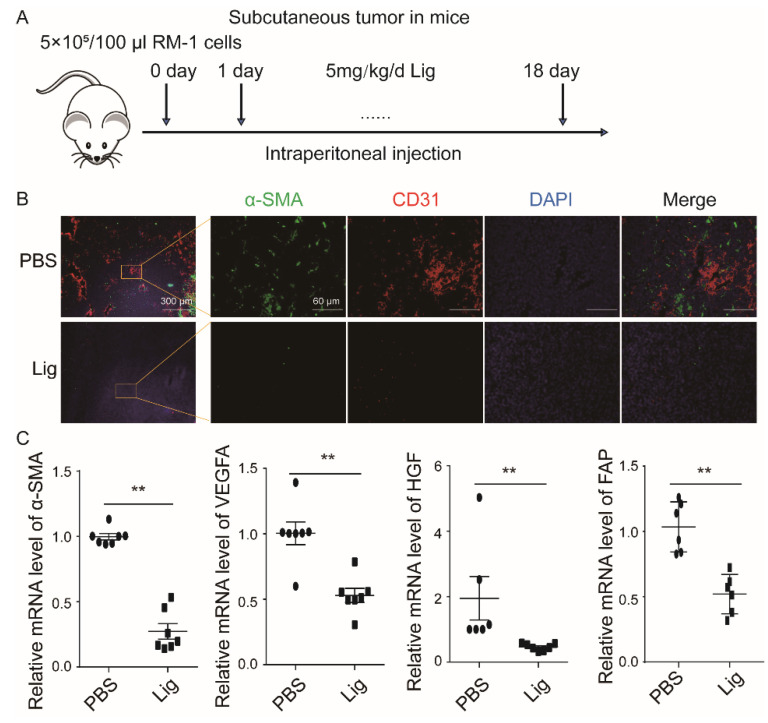
Ligustilide reduces blood vessel density in prostate cancer tissue. A total of 5 × 10^5^ RM-1 cells were injected subcutaneously into C57BL/6 mice to establish a tumor-bearing model treated with intraperitoneal injection of 100 µL of ligustilide (5 mg/kg) or PBS once daily from the second day after tumor bearing to the 18th day. (**A**). Immunohistochemical staining of tumor tissue showed that ligustilide downregulated α-SMA and CD31 levels. (**B**). Ligustilide-decreased mRNA expression levels of α-SMA, VEGFA, HGF and FAP in RT-qPCR analyses. (**C**). Data were normalized against the control and plotted as percentage differences. ** *p* < 0.01.

**Figure 2 cancers-14-02406-f002:**
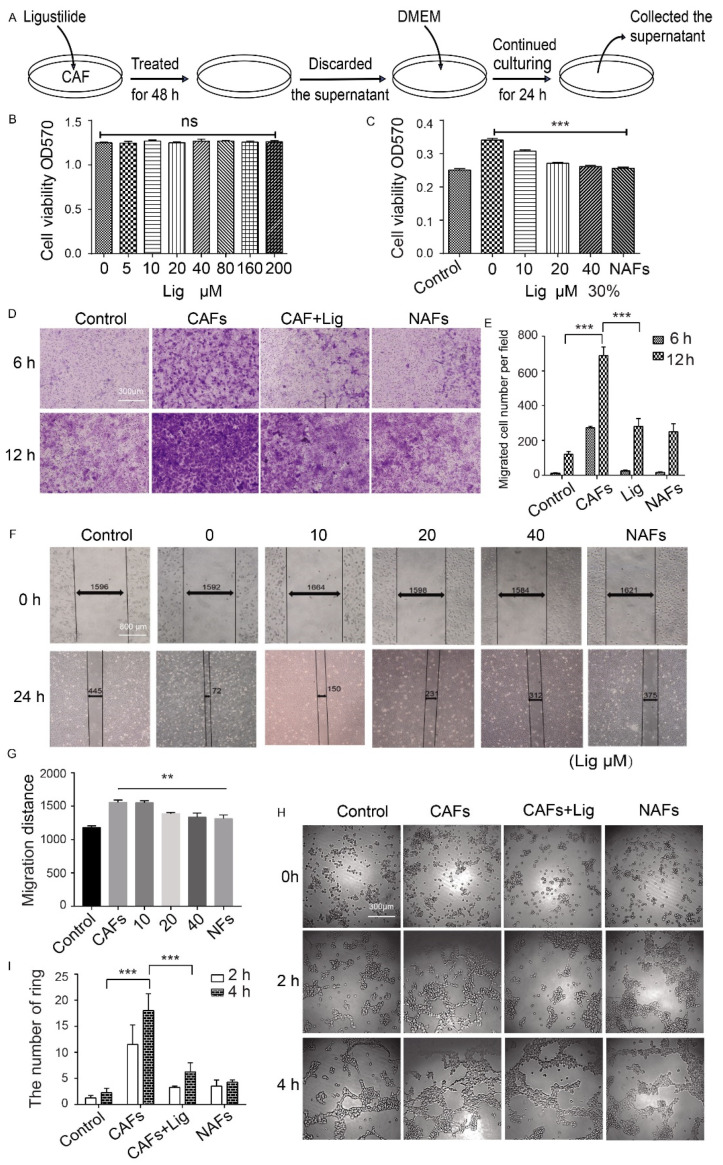
Ligustilide significantly inhibits the pro-angiogenesis effect of CAF supernatant. CAFs were pretreated with ligustilide for 48 h, the medium was replaced with fresh DMEM and the supernatant was collected over 24 h (**A**). The MTT assay shows the proliferation of HUVECs treated with different concentrations of ligustilide for 48 h (**B**). The effect of CAF supernatant pretreated with different concentrations of ligustilide (0, 10, 20 and 40 μM) on HUVEC proliferation using MTT (**C**), migration in Transwell culture systems and statistical results of HUVECs that migrated into the lower chamber of the Transwell (**D**,**E**). Migration in scratch assay and statistical results (**F**,**G**). Tube-like structure formation and statistical results (**H**,**I**). *** p* < 0.01 and **** p* < 0.001, ns: not significant.

**Figure 3 cancers-14-02406-f003:**
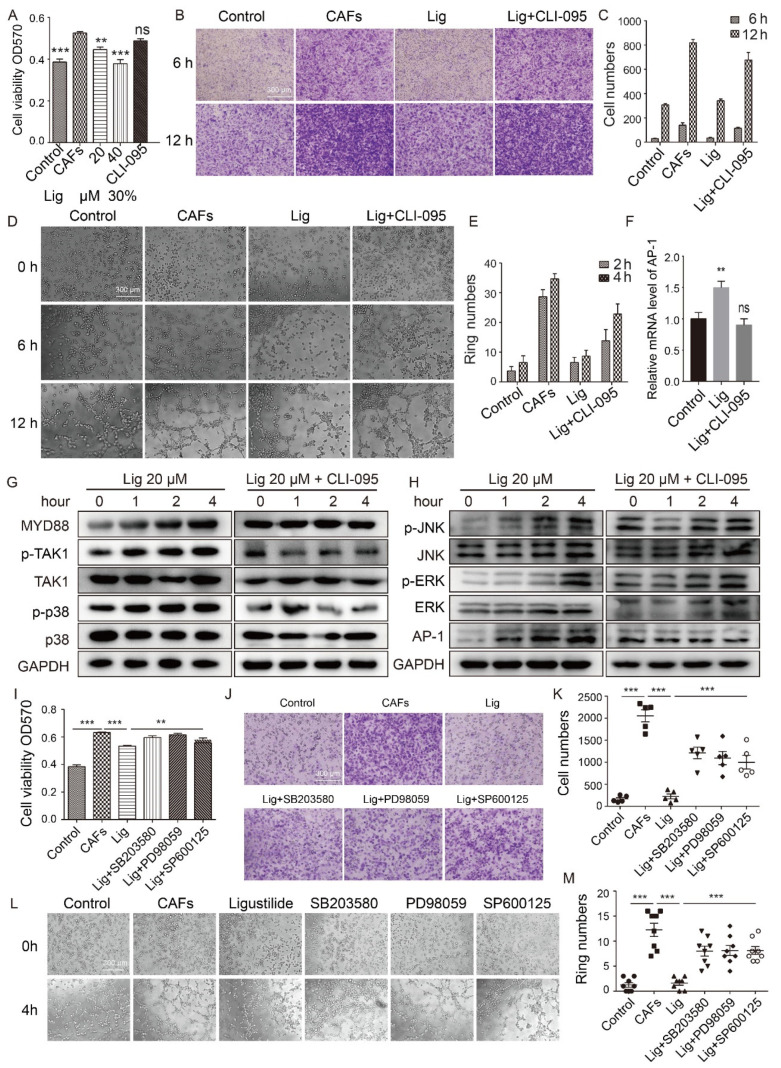
Ligustilide inhibits the pro-angiogenic effect of CAFs via the TLR4-AP-1 signaling pathway. MTT detection of CAF supernatants treated with different concentrations of ligustilide (0, 20 or 40 μM) or CLI-095 (TLR4 inhibitor, 10 nM) pretreatment then treated with ligustilide (40 μM) in the supernatants of CAFs to induce HUVEC proliferation (**A**). The supernatant of CAFs, the supernatant of CAFs treated with ligustilide or the supernatant of CAFs treated with ligustilide after pretreatment with CLI-095 was photographed under a microscope (×40) to record the migration (**B**,**C**) and tube formation (**D**,**E**) of HUVECs. PCR detection of AP-1 mRNA levels in CAFs treated with ligustilide or ligustilide after pretreatment with CLI-095 (**F**). Western blot detection of MYD88, *p*-TAK1, *p*-p38, *p*-JNK, *p*-ERK and AP1 protein levels in CAFs treated with ligustilide or ligustilide after pretreatment with CLI-095 (**G**,**H**). GAPDH was used as a loading control. SB203580 (p38 inhibitor, 30 μM), PD98059 (ERK inhibitor, 20 μM) and SP600125 (JNK inhibitor, 5 μM) inhibited the effect of ligustilide on HUVEC proliferation (**I**), migration (**J**,**K**) and tube formation (**L**,**M**). PBS was used as the control. Each dataset represents the mean ± SD of at least three independent experiments. *** p* < 0.01 and **** p* < 0.001, ns: not significant. The original blots could be found in Appendix A.

**Figure 4 cancers-14-02406-f004:**
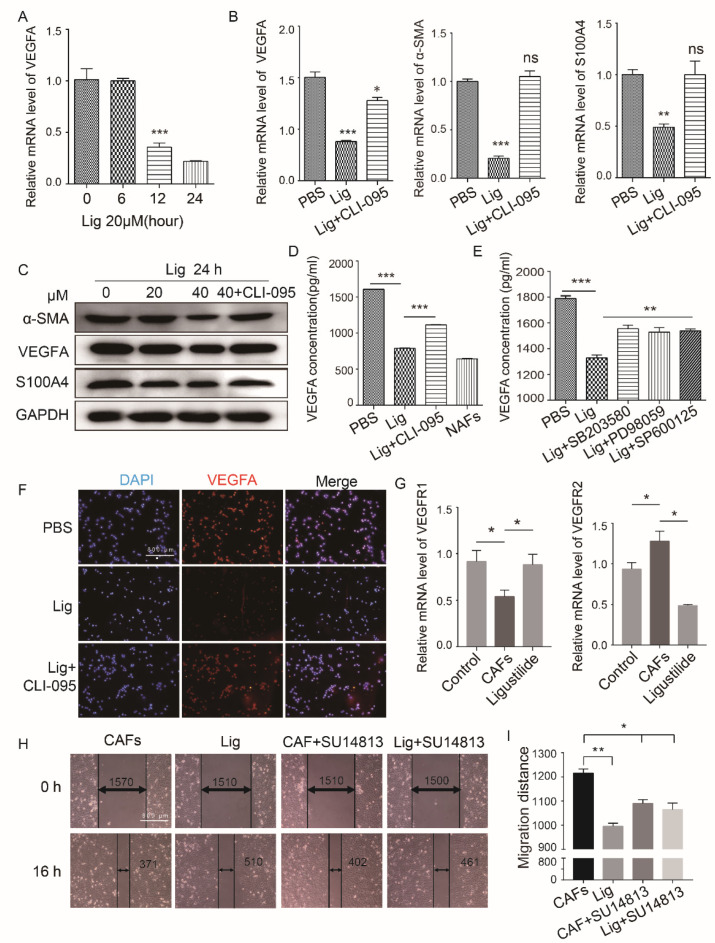
Ligustilide downregulated the expression level of VEGFA in CAFs via the TLR4-AP-1 signaling pathway. Detection of VEGFA mRNA levels in CAFs treated with ligustilide (20 μM) for 0, 6, 12 or 24 h using PCR (**A**). The mRNA levels of VEGFA, α-SMA and S100A4 in ligustilide-treated (20 μM) CAFs and CLI-095 (10 nM) CAFs pretreated with ligustilide were detected using PCR (**B**). The protein levels of α-SMA, VEGFA and S100A4 in these cells were detected using Western blotting (**C**). The amount of VEGFA in the culture supernatant of these CAFs was detected using ELISA (**D**). PBS-treated CAFs and NAFs were used as controls. ELISA was used to detect the amount of VEGFA in the culture supernatant of ligustilide-treated CAFs and SB203580-, PD98059- or SP600125-pretreated CAFs and ligustilide-treated CAFs (**E**). The expression of VEGFA in CAFs was detected using immunofluorescence (**F**). CAF supernatant decreased VEGFR1 and increased VEGFR2 at the mRNA level in HUVECs, and ligustilide reversed this effect (**G**). SU14813 partially reduced the suppressive effect of ligustilide-treated CAF supernatant on HUVEC migration (**H**,**I**). SU14813 is a VEGFR1/2 inhibitor and represents HUVECs pretreated with 50 nM SU14813 for 12 h. ** p* <0.05, *** p* < 0.01 and **** p* < 0.001, ns: not significant. The original blots could be found in Appendix A.

**Figure 5 cancers-14-02406-f005:**
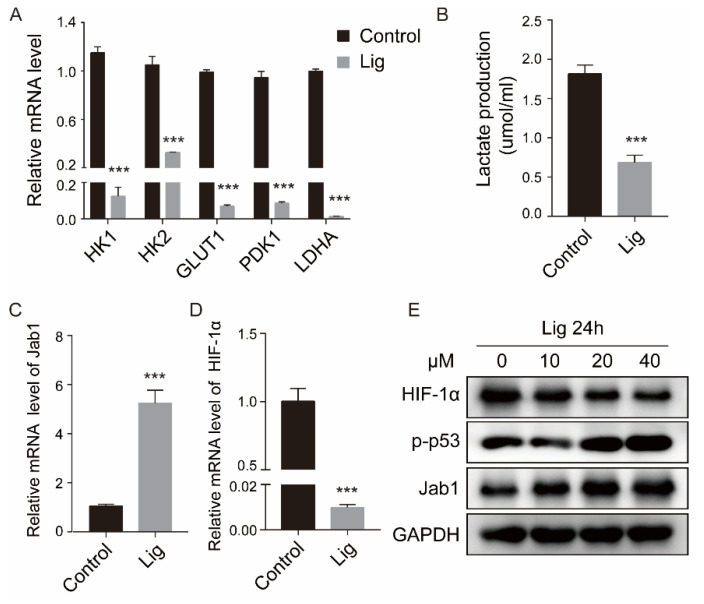
Ligustilide inhibited glycolysis and HIF-1 expression in CAFs. PCR analyses showed the downregulation of HK1, HK2, GLUT1, PDK1 and LDHA, which are involved in glycolysis, in CAFs treated with 40 μM ligustilide for 24 h (**A**). Lactate production was decreased in ligustilide-treated CAFs (**B**). Upregulation of Jab1 (**C**) and downregulation of HIF-1 (**D**) at the mRNA level were detected in ligustilide-treated CAFs. Western blot analyses showed downregulation of HIF-1 and upregulation of p53 and Jab1 in ligustilide-treated CAFs (**E**). **** p* < 0.001. The original blots could be found in Appendix A.

**Figure 6 cancers-14-02406-f006:**
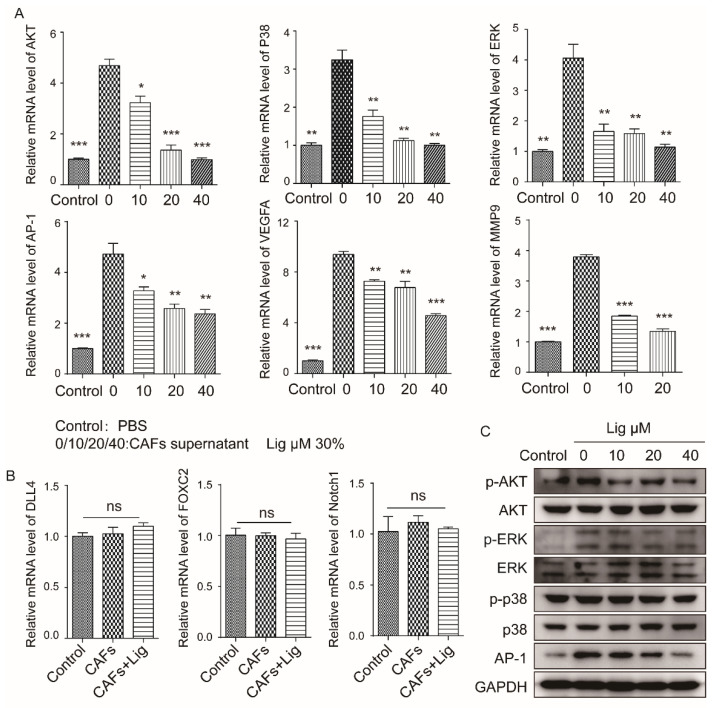
Ligustilide attenuated the signaling pathway involved in the proliferation of HUVECs induced by CAF supernatant. Ligustilide-treated (0, 10, 20 or 40 μM) CAF supernatant suppressed the upregulation of AKT, p38, ERK, AP-1, VEGFA and MMP9 induced by CAF supernatant at the mRNA level in HUVECs (**A**). Ligustilide-treated CAF supernatant had no effect on DLL4, FOXC2 or Notch1 compared with CAF supernatant at the mRNA level in HUVECs (**B**). Ligustilide-treated CAF supernatants decreased *p*-AKT, AKT, *p*-ERK, ERK, *p*-p38, p38 and AP1 protein levels induced by CAF supernatant (**C**). Control represents HUVECs treated with PBS. ** p* < 0.05, *** p* < 0.01 and **** p* < 0.001, ns: not significant. The original blots could be found in Appendix A.

**Figure 7 cancers-14-02406-f007:**
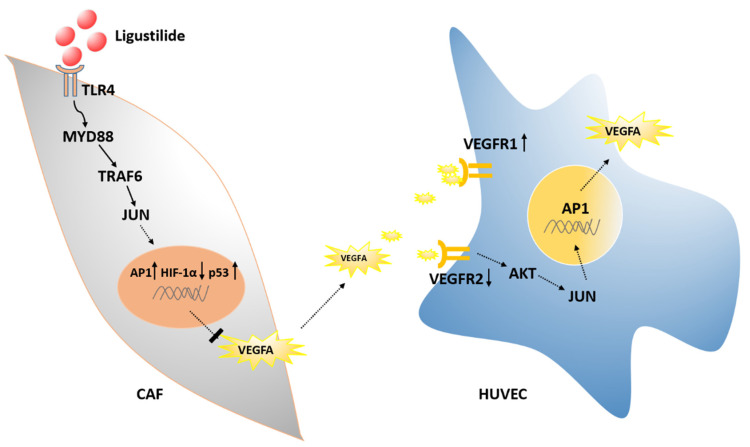
Ligustilide inhibits tumor angiogenesis by downregulating VEGFA secretion from CAFs in prostate cancer via TLR4.

## Data Availability

The data presented in this study are available on request from the corresponding author.

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
