# Peer review of "Ligustilide Inhibits Tumor Angiogenesis by Downregulating VEGFA Secretion from Cancer-Associated Fibroblasts in Prostate Cancer via TLR4"

_cancers, 2022, doi:10.3390/cancers14102406_

Round 1
Reviewer 1 Report
The authors found that ligustilide, an extract of the Chinese herbs Angelica and Chuanxiong, down-regulated the VEGFA level in prostate cancer related fibroblasts (CAFs) via the TLR4-ERK/JNK/p38 signaling pathway and the decrease of HIF-1 expression, and inhibited the promotion of angiogenesis in prostate cancer cells. This study is interesting and well designed, and provides a new strategy for the anti-tumor effect of natural active molecules.
Minor points:
- 3L is not clear. We can not see the ring structure.
- The authors should provide the statistical analysis information in Fig. 3K and 3M.
- VEGFR can activate the phosphorylation of AKT and other downstream molecules through tyrosine kinase. Why the expression of AKT and ERK was induced by VEGFR activation in HUVECs? Please explain. And the expression of total protein of AKT and ERK was not detected in these HUVECs.
- The decrease of HIF-1 by ligustilide is not mention if the “Abstract” section. This should be an important mechanism of VEGFA inhibition too.
Author Response
Review1:
The authors found that ligustilide, an extract of the Chinese herbs Angelica and Chuanxiong, down-regulated the VEGFA level in prostate cancer related fibroblasts (CAFs) via the TLR4-ERK/JNK/p38 signaling pathway and the decrease of HIF-1 expression, and inhibited the promotion of angiogenesis in prostate cancer cells. This study is interesting and well designed, and provides a new strategy for the anti-tumor effect of natural active molecules.
Minor points:
- 3L is not clear. We can not see the ring structure.
Answer: We have improved the quality of Figure 3L.
- The authors should provide the statistical analysis information in Fig. 3K and 3M.
Answer: We have provided the statistical analysis information in Fig. 3K and 3M.
- VEGFR can activate the phosphorylation of AKT and other downstream molecules through tyrosine kinase. Why the expression of AKT and ERK was induced by VEGFR activation in HUVECs? Please explain. And the expression of total protein of AKT and ERK was not detected in these HUVECs.
Answer: AKT and ERK are the signal pathway in the cell proliferation, and VEGFR is the proliferation receptor of HUVECs. The expression of total protein of AKT and ERK were detected in these HUVECs (Fig. 6C).
- The decrease of HIF-1 by ligustilide is not mention if the “Abstract” section. This should be an important mechanism of VEGFA inhibition too.
Answer: The decrease of HIF-1 by ligustilide had been mentioned in “Abstract”. “In addition, Ligustilide inhibited glycolysis and HIF-1 expression in CAFs.”
Review1:
The authors found that ligustilide, an extract of the Chinese herbs Angelica and Chuanxiong, down-regulated the VEGFA level in prostate cancer related fibroblasts (CAFs) via the TLR4-ERK/JNK/p38 signaling pathway and the decrease of HIF-1 expression, and inhibited the promotion of angiogenesis in prostate cancer cells. This study is interesting and well designed, and provides a new strategy for the anti-tumor effect of natural active molecules.
Minor points:
- 3L is not clear. We can not see the ring structure.
Answer: We have improved the quality of Figure 3L.
- The authors should provide the statistical analysis information in Fig. 3K and 3M.
Answer: We have provided the statistical analysis information in Fig. 3K and 3M.
- VEGFR can activate the phosphorylation of AKT and other downstream molecules through tyrosine kinase. Why the expression of AKT and ERK was induced by VEGFR activation in HUVECs? Please explain. And the expression of total protein of AKT and ERK was not detected in these HUVECs.
Answer: AKT and ERK are the signal pathway in the cell proliferation, and VEGFR is the proliferation receptor of HUVECs. The expression of total protein of AKT and ERK were detected in these HUVECs (Fig. 6C).
- The decrease of HIF-1 by ligustilide is not mention if the “Abstract” section. This should be an important mechanism of VEGFA inhibition too.
Answer: The decrease of HIF-1 by ligustilide had been mentioned in “Abstract”. “In addition, Ligustilide inhibited glycolysis and HIF-1 expression in CAFs.”

Reviewer 2 Report
The study shows that Ligustilide decreases SMA+ and CD31+ cells in a subcutaneous allograft model of prostate cancer (RM-1). In vitro experiments show that Ligustilide treated CAF have a reduced ability to promote HUVEC migration and tube formation. In vitro drug studies support a role for TLR4-induced signaling and VEGFA expression. Overall, the study supports a model that Ligustilide restricts angiogenesis by attenuating VEGFA expression vis TLR4 in CAFs.
Concerns
- It would be helpful to show the size of the tumors in Figure 1 that were treated with vehicle or Ligustilide. Does Ligustilide have the same impact on CAFs and endothelial cells if the tumors have been fully established prior to treatment? It would be helpful to show a larger area of the tumors for SMA and CD31 by IHC.
- In Figure 2, only one CAF cell line was used and only NAFs from one patient were used. These assays would have more rigor if additional cell lines were used for the assays.
- It would be informative to see TLR4 expression by western in CAFs.
- How Ligustilide mechanistically activates TLR4 in CAFs was not shown.
- How Ligustilde impacts other cell types in the tumor microenvironment that express TLR4 was not shown.
Minor Concerns
- HUVEC tube formation in the photos was difficult to see.
- Editing for grammar and accuracy is suggested.
Author Response
Review2:
The study shows that Ligustilide decreases SMA+ and CD31+ cells in a subcutaneous allograft model of prostate cancer (RM-1). In vitro experiments show that Ligustilide treated CAF have a reduced ability to promote HUVEC migration and tube formation. In vitro drug studies support a role for TLR4-induced signaling and VEGFA expression. Overall, the study supports a model that Ligustilide restricts angiogenesis by attenuating VEGFA expression vis TLR4 in CAFs.
Concerns
- It would be helpful to show the size of the tumors in Figure 1 that were treated with vehicle or Ligustilide. Does Ligustilide have the same impact on CAFs and endothelial cells if the tumors have been fully established prior to treatment? It would be helpful to show a larger area of the tumors for SMA and CD31 by IHC.
Answer: The size of the tumors in Figure 1 has been reported in previous manuscript as reference 21 [Ma Jing, et al. Food and Chemical Toxicology, 2020, 135: 110991]. It is unknown that the impact of ligustilide CAFs and endothelial cells if the tumors have been fully established prior to treatment. A larger area of the tumors for SMA and CD31 by IHC are showed as Fig. 1B:
- In Figure 2, only one CAF cell line was used and only NAFs from one patient were used. These assays would have more rigor if additional cell lines were used for the assays.
Answer: We have added additional cell lines of CAFs (CT26-CAFs) [Rong L, et al. Oncotarget (2017) 8:97231–45.] were used for the assays. CT26-CAFs were constructed from mouse embryonic fibroblasts (MEFs). MEFs were transfected with hTERT retrovirus containing a puromycin resistance gene. Immortalized MEFs with puromycin-resistance were co-injected with CT-26 cells to build tumor bearing mouse model. Tumor was dissociated after 10 days tumor growth and 2 mg/ml puromycin were used as selection. MEFs upon co-injection with tumor cells mainly differentiated into CAFs. Ligustilide also decreased the VEGFA level in CT26-CAF, which is similar to prostate-CAF (Fig. S5C).
- It would be informative to see TLR4 expression by western in CAFs.
Answer: We had detected the TLR4 expression by western in CAFs as following and added it in supplemental materials (Fig. S3C).
- How Ligustilide mechanistically activates TLR4 in CAFs was not shown.
Answer: It is unknown that how Ligustilide mechanistically activates TLR4 in CAFs. In further research, it is an important question to study.
- How Ligustilide impacts other cell types in the tumor microenvironment that express TLR4 was not shown.
Answer: We have added myeloid derived suppressor cells (MDSCs) line (MSC2) which are other immune suppressor cell types that express TLR4 by western blot (Fig. S5D). The impact of ligustilide on VEGFA level had been detected.
Minor Concerns
- HUVEC tube formation in the photos was difficult to see.
Answer: We have improved the photos of HUVEC tube formation (Fig. 3D and Fig. 3L)
- Editing for grammar and accuracy is suggested.
Answer: Thanks for your suggestion, and the editing for grammar and accuracy are provided by American Journal Experts, LLC.

Reviewer 3 Report
The current article from Ma and Chen et.al., titled "Ligustilide inhibits tumor angiogenesis by downregulating 2 VEGFA secretion from cancer-associated fibroblasts in prostate 3 cancer via TLR4", authors have tried to show interplay between CAFs and angiogenesis which has been reported by several other groups. Overall the manuscript addresses one of the most important event that are being addressed in the tumor microenvironment field and this article will certainly establish the current hypothesis. However proper care has not been taken while preparing the manuscript; several controls are missing in western blots and most of the images are hard to read and interpret. The manuscript is not well written and has introduced too many variables. I would recommend authors to reorganize the manuscript so that it can be easily understood.
- My main concern is with figure 1, authors have used subcutaneous tumor model, and have observed very high CAFs, this is not a valid claim. When a tumor is implanted subcutaneously most of the fibroblasts are dermal fibroblasts and there is a little to no infiltration of CAFs in to the tumor. However orthotopic tumors may have slightly higher CAFs. I would suggest authors provide IHC images and quantification for αSMA and FAP which would be a more representative of actual levels of stromal cells.
- One of the more important experiments I would request authors to perform, since authors have mentioned that CAFs secreted growth factors are responsible, if authors can validate the tube formation and migration assay with Heat inactivated CAF media and add the specific factors that further strengthen the said hypothesis.
- In figure 3G and 3H most of the blots are not clean and lack proper controls.
- My main concern is with Figure 6, authors have demonstrated that upon Ligustilide treatment several kinase levels have decreased, this could be mainly due to cell death but not due to decrease in the expression of the kinase levels for AKT, p38, and ERK, this really does not reflect the actual phenomenon happening. This is something not possible to explain with the current data. I would suggest authors to run western blots for these samples.
- In figure 6C specifically authors have observed the decrease of p-AKT and p-P38, however the controls are missing in this set of blots, authors need to have blots for AKT, P38, p-ERK and ERK.
- It is extremely difficult to see the tube formation assay, migration or scratch assays as most of the images are blurry and lack the journals standards and are extremely difficult to read and interpret.
- Authors have claimed the role from secreted VEGF is responsible for the effects on endothelial and HUVECS, but authors have not used VEGF functional blocking antibody (Bevacizumab) to block this event, this is the most important control the authors have missed in the entire manuscript.
- Scale bars are missing on almost all the images.
Author Response
Review3:
The current article from Ma and Chen et.al., titled "Ligustilide inhibits tumor angiogenesis by downregulating 2 VEGFA secretion from cancer-associated fibroblasts in prostate 3 cancer via TLR4", authors have tried to show interplay between CAFs and angiogenesis which has been reported by several other groups. Overall the manuscript addresses one of the most important event that are being addressed in the tumor microenvironment field and this article will certainly establish the current hypothesis. However proper care has not been taken while preparing the manuscript; several controls are missing in western blots and most of the images are hard to read and interpret. The manuscript is not well written and has introduced too many variables. I would recommend authors to reorganize the manuscript so that it can be easily understood.
- My main concern is with figure 1, authors have used subcutaneous tumor model, and have observed very high CAFs, this is not a valid claim. When a tumor is implanted subcutaneously most of the fibroblasts are dermal fibroblasts and there is a little to no infiltration of CAFs into the tumor. However orthotopic tumors may have slightly higher CAFs. I would suggest authors provide IHC images and quantification for αSMA and FAP which would be a more representative of actual levels of stromal cells.
Answer: We have added IHC images with a large scope (Fig. 1B), and quantification for FAP by qPCR (Fig. 1C).
- One of the more important experiments I would request authors to perform, since authors have mentioned that CAFs secreted growth factors are responsible, if authors can validate the tube formation and migration assay with Heat inactivated CAF media and add the specific factors that further strengthen the said hypothesis.
Answer: We have validated the tube formation and migration assay with heat inactivated CAF media and add the specific factors (VEGFA) (Fig. S6E and Fig. S6F).
- In figure 3G and 3H most of the blots are not clean and lack proper controls.
Answer: We have improved the figure 3G and 3H.
- My main concern is with Figure 6, authors have demonstrated that upon Ligustilide treatment several kinase levels have decreased, this could be mainly due to cell death but not due to decrease in the expression of the kinase levels for AKT, p38, and ERK, this really does not reflect the actual phenomenon happening. This is something not possible to explain with the current data. I would suggest authors to run western blots for these samples.
Answer: We have run western blots for these samples to detect the expression of the AKT, P38 and ERK (Fig. 6C).
- In figure 6C specifically authors have observed the decrease of p-AKT and p-P38, however the controls are missing in this set of blots, authors need to have blots for AKT, P38, p-ERK and ERK.
Answer: In figure 6C, we have added the detection of AKT, p38, p-ERK and ERK.
- It is extremely difficult to see the tube formation assay, migration or scratch assays as most of the images are blurry and lack the journals standards and are extremely difficult to read and interpret.
Answer: We have improved the tube formation assay (Fig. 2H, Fig. 3D, Fig. 3L, Fig. S6E and Fig. s6F), migration (Fig. 2D, Fig 3B, Fig. S2B, Fig. S6A and Fig. S6B) and scratch assays (Fig. 2F, Fig. 4H, Fig. S6C and Fig. S6D).
- Authors have claimed the role from secreted VEGF is responsible for the effects on endothelial and HUVECS, but authors have not used VEGF functional blocking antibody (Bevacizumab) to block this event, this is the most important control the authors have missed in the entire manuscript.
Answer: We added VEGF functional blocking antibody (Bevacizumab) to block this event (Fig. S6E and Fig. S6F).
- Scale bars are missing on almost all the images.
Answer: We have added scale bars on the images (Fig. 1B, Fig. 4F, Fig. S4).

Reviewer 4 Report
The manuscript titled "Ligustilide inhibits tumor angiogenesis by downregulating 2 VEGFA secretion from cancer-associated fibroblasts in prostate 3 cancer via TLR4" focus on investigating Ligustilide anti-cancer mechanism. The authors well designed the experiments to study and validate Ligustilide mechanism using different biochemical techniques. However, it was not clear how the authors drawn conclusion to summarize Figure 7.
Most of the figures have too many panels, it would be nice to move some of the panels to supplementary information.
Few other comments:
In simple summary, too many acronyms are included, it would be nice to write in layman's language.
Abbreviations are repeated several times in text and some times it is not clear. For ex. line 55, cancer-associated fibroblasts (CAFs) and in line 91, Prostate cancer-associated fibroblasts (CAFs), it was not clear what stands for CAFs.
Conclusion section is too short.
Author Response
Review4:
The manuscript titled "Ligustilide inhibits tumor angiogenesis by downregulating 2 VEGFA secretion from cancer-associated fibroblasts in prostate 3 cancer via TLR4" focus on investigating Ligustilide anti-cancer mechanism. The authors well designed the experiments to study and validate Ligustilide mechanism using different biochemical techniques. However, it was not clear how the authors drawn conclusion to summarize Figure 7.
Most of the figures have too many panels, it would be nice to move some of the panels to supplementary information.
Answer: These figures were supported the results from different aspect or different methods, and we applied to keep the panels.
Few other comments:
In simple summary, too many acronyms are included, it would be nice to write in layman's language.
Answer: We have added the whole name of these acronyms, such as vascular endothelial growth factor A (VEGFA), cancer associated fibroblasts (CAFs) and Toll-like receptor4 (TLR4) in simple summary.
Abbreviations are repeated several times in text and some times it is not clear. For ex. line 55, cancer-associated fibroblasts (CAFs) and in line 91, Prostate cancer-associated fibroblasts (CAFs), it was not clear what stands for CAFs.
Answer: Cancer-associated fibroblasts stands for CAFs. Line 91, Prostate cancer-associated fibroblasts (CAFs) has changed to (P-CAFs).
Conclusion section is too short.
Answer: We added the content of “The effect related to the down-regulation of HIF-1α.” in conclusion.

Round 2
Reviewer 3 Report
The authors have addressed all the comments, I believe the manuscript is significantly improved.
Reviewer 4 Report
The authors adequately addressed my comments to improve quality of the manuscript. I recommend this manuscript for publication.